# Assessing Sustainability in the Shipbuilding Supply Chain 4.0: A Systematic Review

**Magdalena Ramirez-Peña** [1,*] **, Francisco J. Abad Fraga** [2] **, Jorge Salguero** [1] **and Moises Batista** [1,*]

1   Department of Mechanical Engineering and Industrial Design, School of Engineering, University of Cádiz, E-11519 Puerto Real-Cádiz, Spain; jorge.salguero@uca.es
2   Navantia S.A., Astillero Bahía de Cádiz, Polígono Astilleros s/n, E-11519 Puerto Real-Cádiz, Spain; fabad@navantia.es
*   Correspondence: magdalena.ramirez@uca.es (M.R.-P.); moises.batista@uca.es (M.B.)

**Abstract:** The supply chain is currently taking on a very important role in organizations seeking to improve the competitiveness and profitability of the company. Its transversal character mainly places it in an unbeatable position to achieve this role. This article, through a study of each of the key enabling technologies of Industry 4.0, aims to obtain a general overview of the current state of the art in shipbuilding adapted to these technologies. To do so, a systematic review of what the scientific community says is carried out, dividing each of the technologies into different categories. In addition, the global vision of countries interested in each of the enabling technologies is also studied. Both studies present a general vision to the companies of the concerns of the scientific community, thus encouraging research on the subject that is focused on the sustainability of the shipbuilding supply chain.

**Keywords:** sustainability; supply chain; shipbuilding; key enabling technologies; industry 4.0

---

## 1. Introduction

The existing flow of materials and information within an organization is defined as the supply chain, and it goes from the suppliers of raw materials to the consumer of the final product [1]. In addition, the Council of Supply Chain Management Professionals also assigns it the role of integrator among all actors involved. The evolution of the supply chain has reached the point where it is considered a strategic concept within the business model of companies [2]. This strategic tool, with a multidisciplinary and transversal character, affects the three strategic levels that are distinguished in the organizations. The first strategic level defines where the organization is framed and the market in which it competes. The second strategic level defines how it will compete and the third, functional strategy as put into practice within each area that makes up the company [3].

Based on this transversality, which is assigned to the supply chain, the models that can be most interesting to follow are studied. The techniques and practices provided by lean manufacturing are fully applicable to the supply chain as it is considered a network of small businesses that becomes a network of small independent companies, which must be coordinated in the best possible way. A lean supply chain supports collaborative relationships based on mutual trust between suppliers, develops programs to give them technical support, establishes open door policies, and promotes participation from the first stages of working as a team in the search for solutions [4].

The agile contribution to the supply chain allows it to respond to the continuous changes that exist in the market by establishing new competencies. It is based on a dynamic structure management providing information visibility [5]. There is also the supply chain defined as a combination of both

paradigms, known as leagile [6]. Most studies focus on the supply chain from a financial risk aspect, which has led to new studies from a social perspective, hence creating a new model [7]. In addition, it is known at the outset how many interruptions a supply chain will be subjected to, and, knowing this, the most appropriate approach would be to try to prevent and properly manage the changes in status that the supply chain may be subjected to. Resilience offers this contribution [8].

Special mention is made of sustainability. Sustainability and green—there are different terms that refer to this concept but, although there are nuances, the objective is the same. It must be understood that the sustainable supply chain concerns the creation of economic, environmental, and social considerations—in other words, integrating the environment into the management of the supply chain [9].

For the shipbuilding sector, sustainability carries a very important weight. To be able to integrate the environmental dimension into all the operations carried out within a shipyard has been a matter of vital importance in the last years. Therefore, one of the advantages of such integration into the supply chain will give the company a certain competitive advantage, for example, with regard to improving energy efficiency [10,11].

Considered also are the efforts that have been lately directed towards its adaptation to the requirements marked by Industry 4.0 in order not to be left behind in the market, this being the only way to survive in such a competitive market in a sector. That is why it is proposed to improve management by using a tool as useful in this sense as the systematic review for companies. For companies, used to using business articles, the contribution provided by a scientific study is beneficial for decision-making [12].

The term Industry 4.0 originated in Germany, where Kagermann, Lukas and Wahlster based their industrialization proposal on nine high technologies, in addition to establishing strategies for their implementation, which were later known as Key Enabling Technologies (KETs) [13,14]. Some authors have varied these technologies, adapting them as best suits their sector. In shipbuilding, there have been few contributions and differences with respect to overall adaptation [15]. In our study, those described in a conceptual model developed specifically for shipbuilding are considered [16]. These studies have even allowed defining an index that allows evaluating the state of maturity in the implantation in the company [17]. These technologies will affect the development of new products and services, the business models carried out by organizations, and the supply chain, creating competitive advantage and cost reduction. In order to generate benefits for all stakeholders, Supply Chain 4.0 can define itself as the transformation of the traditional supply chain using enabling technologies [18–20]. This is not the only new dimension of Supply Chain 4.0; it must also be supported by other new dimensions, such as those related to management and capacity supports, process performance requirements, and strategic results. This makes the concept of Supply Chain 4.0 an evolution of the traditional concept which, despite being in its initial period, is in the process of development [18]. This development of Supply Chain 4.0 can be considered as a transformation that includes the incorporation of technologies in addition to the human and environmental dimensions, placing sustainability at the center of improving the company [19]. Furthermore, Industry 4.0 itself helps industries to incorporate actions for the protection and control of the environment by converting supply chains into Sustainable Supply Chains 4.0. The purpose of these Sustainable Supply Chains 4.0 is to plan and project the supply chain itself by taking into account environmental and social concerns besides profits [20].

Therefore, this study aims to give an overview of the state of the art in the shipbuilding adaptation of Industry 4.0. Firstly, it provides an analysis of the interests of countries around the world in Key Enabling Technologies for Industry 4.0. Secondly, it provides a review of studies focusing on making the supply chain sustainable by trying to encourage greater concern about this issue.

## 2. Materials and Methods

In order to carry out the proposed study, the article aimed to follow a systematic review. This systematic review procedure is a tool for both advanced management scholarships and

studies carried out in organizations that wish to improve their management practice. In this way, the management of companies becomes enriched by the contribution of the scientific community, which goes beyond those sources consulted by companies, usually comprising journals more focused on business. This enrichment provides a clear, scientific, and replicable process, and while it does not provide answers, it provides what is known and not known about the question, which are both equally important. The five steps of the systematic review are (1) planning the review, (2) locating studies, (3) assessing contributions, (4) analyzing and synthesizing information, and (5) reporting evidence. These steps will allow getting to know the state of the art of the studied proposal. [12].

There are different possibilities to frame a systematic review, such as PICO (Patient, Intervention, Comparison, Outcomes), SPICE (Stakeholder, Phenomenon of Interest Comparison Evaluation), and CIMO (Context, Intervention, Mechanism, Outcomes). The CIMO logic (Context, Intervention, Mechanism, Outcomes) was adapted to developing the set of propositions of the research sections as being appropriate to the field of management [21], where we could define: Context: Shipbuilding; Intervention: Each of the KETs; Mechanisms: Categories into which the impact on the supply chain has been divided; and Outcomes: Effects of these interventions. Figure 1 shows the methodology followed in the article based on the reviewed literature for the systematic review [12,22].

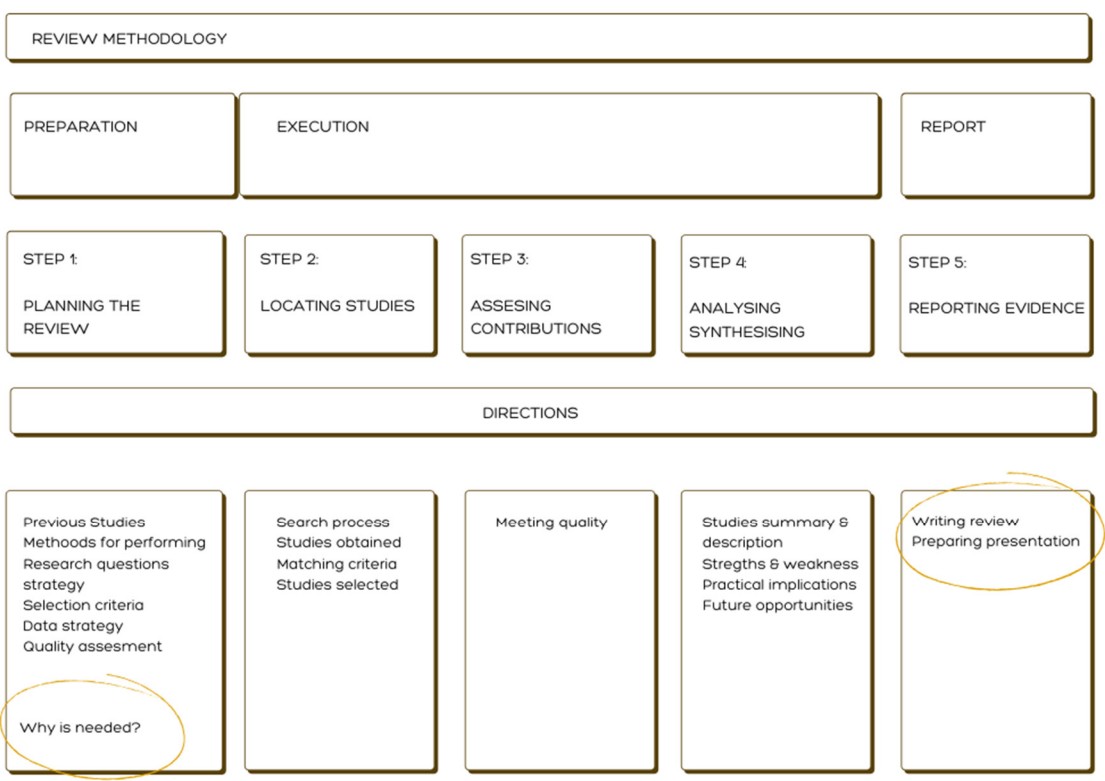

**Figure 1.** Methodology Review. Adapted from [12,22].

During the preparation of the review, the need for it was identified through proposed research questions that were not answered by previous studies. Subsequently, the search strategy was defined, and the selection criteria for data extraction and quality assessment were established. The checklist of Preferred Information Items for Systematic Reviews and Meta-Analyses (PRISMA) [22] was used to provide for the accuracy of the review process.

Regarding the search strategy, different databases were evaluated. We started by carrying out the search with the same argument in several databases and evaluating the answers obtained in each of them, until reaching the conclusion that the Scopus database offered a greater number of contributions, including those provided by the rest of the databases and the possibility of classification with different criteria including impact criteria [23]. No a priori exclusion criteria were made with respect to the time

horizon of the publications since the search arguments already marked recent studies. The established search arguments allowed the intersection of "marine" AND each of the 12 Industry 4.0-enabling technologies for the shipbuilding sector [16]. The term "shipbuilding" was logically the first search argument, although a number of items were not generated that would allow it to be considered an appropriate indicator. Therefore, "shipbuilding" was not considered as a search argument, as it was preferable to establish exclusion criteria extending the search term to "marine" given in this extension, which included the publications generated with shipbuilding.

It was decided that only articles from peer-reviewed journals would be accepted, with the inclusion of a book chapter being an exception. It was the co-authors who decided whether publications were accepted or not, and they debated until agreement was reached. Most of the indexed journals used had a high impact factor between the first two quartiles, considered through the Journal Citation Report (JCR), an appropriate tool for the area in which the study was framed.

Each of the elements studied was collected to be classified according to the context, the intervention, and the mechanism and result relationship. In this way, categories were established that allowed the content of each topic to be analyzed (see Appendix A). In addition, it was decided to carry out a study that would allow us to know which countries were studying what technology, which would allow us to identify the development of each technology at a global level.

## 3. Results

As a first contact and applying the search strategy limited to "supply chain" AND each of the 12 enabling technologies with their most common nomenclatures, shown in Figure 2 as "Search String 1", 680 publications were obtained. However, when the term "marine" was included in the search string, the number of publications decreased considerably to the 284 publications shown in Figure 2 as "Search String 2", which we will study next.

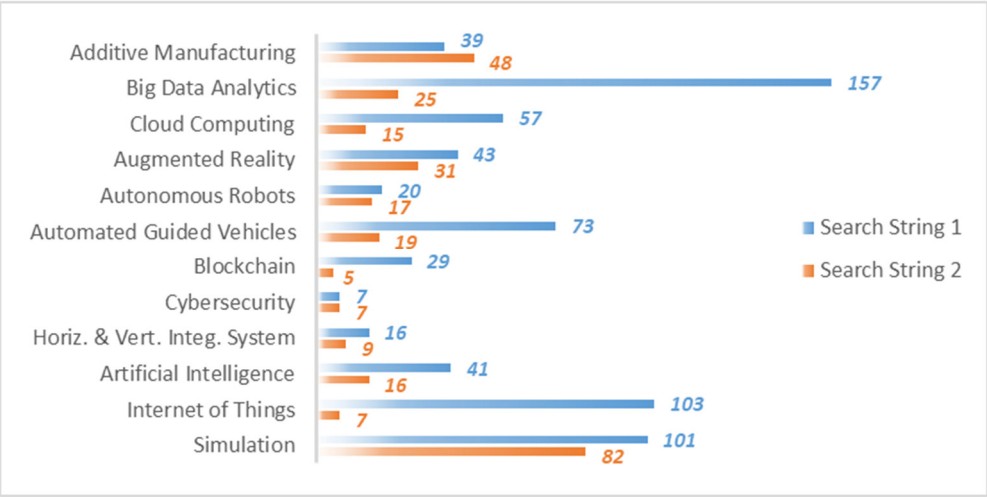

**Figure 2.** Search string comparison.

The second search string was therefore the term "marine" with each of the KETs and filtered by the term "shipbuilding". "Supply chain" was understood as a transversal and driving factor of the shipbuilding industry. At this point it can be stated that many of the publications related to shipbuilding were directly associated with the object of research, without any relation with the term "shipbuilding".

### 3.1. An Overview of the Results

Regarding the country study, Figure 3 shows the distribution of each of the publications grouped by technology. As can be seen, the trend was that these technologies were being studied globally in a general way, however, it was possible to make a breakdown of this.

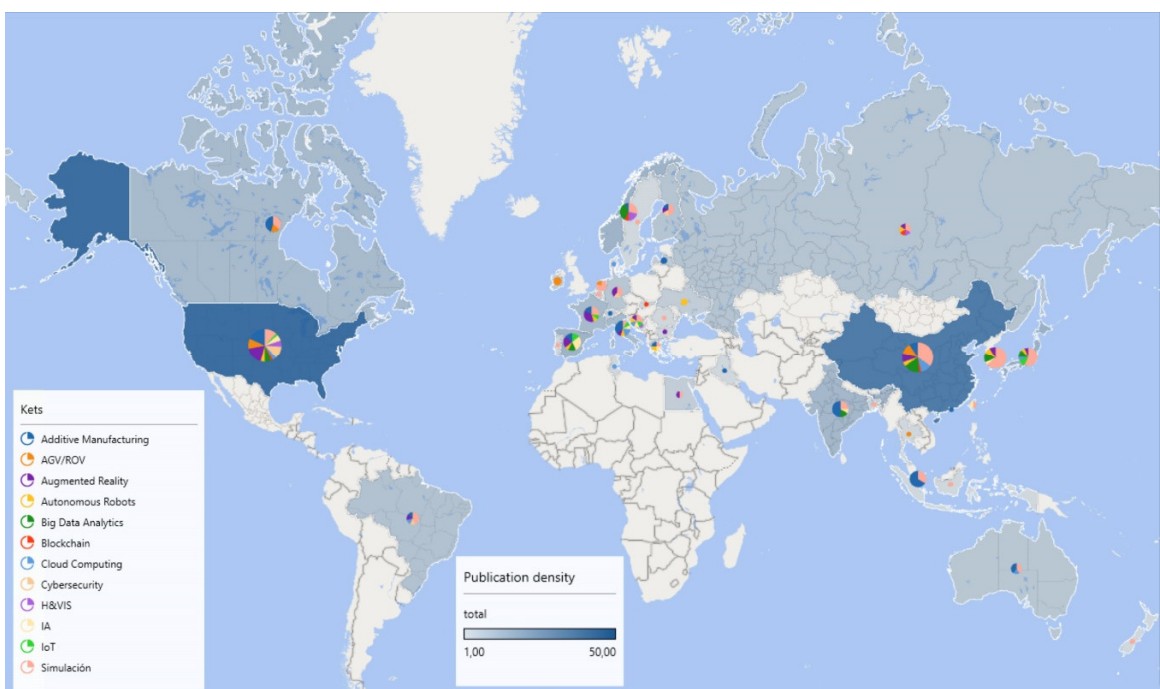

**Figure 3.** Worldwide distribution of technology studies.

The United States was the only country with publications on all enabling technologies, with the highest number on additive manufacturing (9/50). China showed more interest in simulation (15/43) followed by big data (7/43). This behavior was the same for South Korea (simulation: 14/22; big data: 3/22). Neither had publications in cybersecurity, horizontal and vertical integration systems or the Internet of Things. In the same way, England and Japan also showed the highest numbers of publications in simulation; however, Norway had the highest in big data and Spain in artificial intelligence. India, Singapore, and Canada also opted for additive manufacturing, along with Italy, while France was highest in augmented reality. The rest of the contributing countries presented a few publications in each of the technologies, such as the Netherlands with 4/5 in simulation, and, on the other hand, the Internet of things was of interest only to Croatia, outside of the countries with the highest number of publications.

Figure 4 shows the technologies studied by the different countries, as well as the number of publications provided. As stated above, the United States was the only country that studied all the technologies, being the one with the highest number of publications. Of the 44 countries that made up the study, 18 focused on a single technology. Most of the studies dealt with simulation. However, 11 of these countries had interests in different technologies such as cloud computing, automated guided vehicles, augmented reality or autonomous robots.

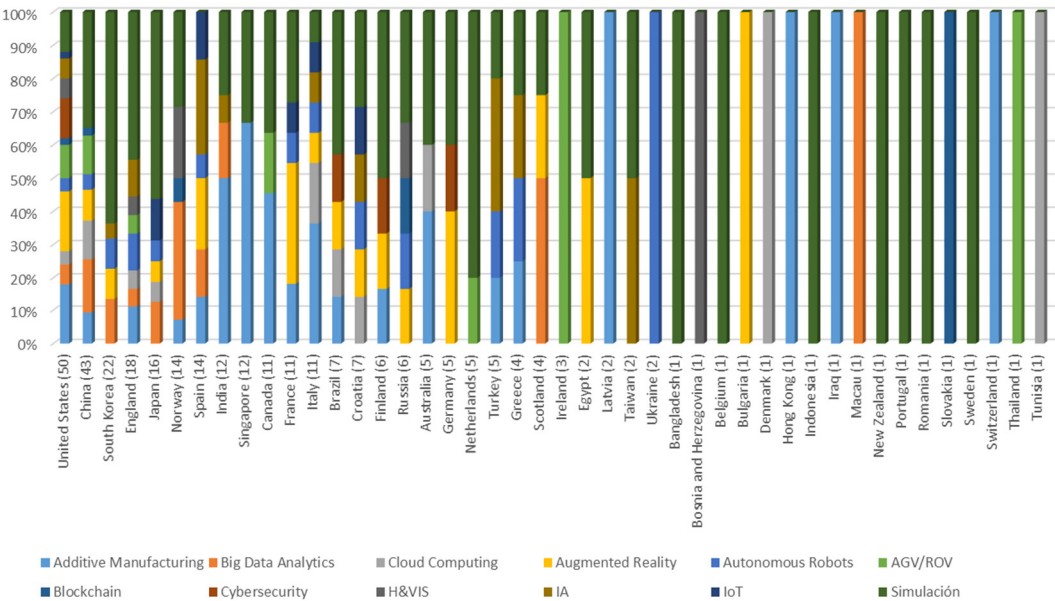

**Figure 4.** Countries' classifications according to the key enabling technologies (KETs) studied.

### 3.2. Analysis of the Key Enabling Technologies

In order to carry out our study and as previously indicated in the methodology, the values of the surveys to be confronted were divided into two groups, one comprising experts of the sector and the other the scientific community. For this second group, using the scientific database, we searched for the up-to-the-minute publications of each of the KETs associated with the shipbuilding supply chain.

Additive manufacturing has been one of the most disruptive manufacturing technologies provided in the context of Industry 4.0. Additive manufacturing can be classified according to the material used, the way it is provided, and the method used to induce consolidation. Based on this, the largest number of published papers were in the general technology group: firstly, the application of technology for metal [24], for composite materials [25], and the simulation of processes for marine components with the intention of inserting the simulation of additive manufacturing in the large-scale shipbuilding environment [26]. Also included were studies on how additive manufacturing benefits the shipbuilding supply chain [27]. Other studies focused on the manufacture of parts using a direct laser-forming technique for the blades of a turbine or to X-band a horn with 3D printing [28], and even the refurbishment of parts outlining the benefits of laser cladding technology for in situ marine crankshaft repairs [29]. There were also articles dealing with the improvement of the properties of manufactured parts, and the elasticity of naval steels [30] on corrosion [31], even going as far as redesigning them for application in additive manufacturing [32].

Big data and analytics enable real-time decision-making through stored data and their evaluation, key to promoting operational excellence by adding value to the company. As applied to shipbuilding, publications were grouped into four groups. In the first group were those aimed at improving processes and systems. One of the most important systems was the navigation system, where solutions are based on ship performance, monitoring and navigation data that improve navigation strategies [33]. There were also processes focused on correlating the sound of the arc with the quality of the welding [34] or studying how the adoption of big data analytics increases the production and productivity of the company and helps it to have more control of its processes [35]. The second group was focused on its potential application motivated by growing concern about climate change [36] and the optimization of energy consumption through the transfer of energy that exists in the hull, the propeller, and the main engine, demonstrating the efficient reduction of energy consumption and $CO_2$ emissions [37]. The third group focused on the improvement of intelligent systems, such as the case of a tool to analyze

data obtained through the Internet of Things (IoT) [38]. Moreover, other publications showed guidance for implementation [39] and sector analysis [40].

Cloud computing improves reaction times and improves production systems through data-based services. There were three different groups, one of which aimed to improve ship behavior in service by analyzing stress, fatigue, fracture [41], its maintenance [42], and improving the safety of ship operations by developing a method of accident analysis with a bridge simulator [43]. The second group focused on improving data management by improving the detection, identification, and ship tracking [44] and their application on ship routing [45]. The third one concerned environmental efficiency through marine engine failure detection to reduce marine pollution [46] and sustainable development [47].

The general application of augmented reality provides workers with information to improve decision-making and work procedures in real time. Applied to shipbuilding, three groups were distinguished, the first one based on a general assessment of the technology of applying augmented reality techniques to learning and for daily management of the marine hydraulic system [48]; on its impact on the sector by considering how it could be applied in order to provide useful and attractive interfaces that allow workers to obtain information about their tasks and to interact with certain elements around them [49]; and on its use as a training tool [50]. A second group focused on its application to simulated naval environments [51], specifically in the fields of navigation, alleviating cognitive load problems for ships [52], safety [53], and maintenance work [51]. A last group aimed at improving the efficiency of systems, for example, through the development of a methodology to match images to different fields of view of the camera and display device by means of coordinate conversion [54].

Autonomous and collaborative robots are born for tackling complex tasks and working as a team with humans. Their main activity in shipbuilding is automated welding, including the welding robots used in the prefabrication of sub-assemblies in production lines, as well as a new high-speed welding process that uses two wires in the welding torch, allowing productivity to be at least doubled [55]. In this case, a rail-running mobile welding robot for the double hull ship structure [56] can be considered for one of the complex tasks mentioned. In addition, there are cleaning and inspection tasks, such as solving the huge environmental and financial problem for the marine industry of marine growth on ships [57], and those corresponding to dimensional control, in particular, the difficulty of measuring marine propellers [58]. The improvement of systems efficiency through simulation is again present in this technology [59], as well as the study of the technology from a general perspective.

Autonomous vehicles was divided into two initial categories, one focused on improving the systems and the other on the applications derived from their use. In the first group, distinctions were made between surface vehicles by studying the propulsion topologies of ships with mechanical, electrical, and hybrid propulsion and energy supply systems and demonstrating that hybrid architectures with advanced control strategies can reduce fuel consumption and emissions, improve noise, maintainability, maneuverability and comfort [60] of underwater vehicles, integrating the obstacle detection and analysis capabilities [61]. Within the applications of their use, there are autonomous vehicles dedicated to Inspection–Maintenance–Repair, so they are called IMR vehicles. Therefore, their use is more focused on inspection by incorporating the Smart Loop Management System (STMS) [62]; others focus on maintenance [63] and on repair, which allows engineers and marine operators to assess the risks associated with certain tasks, such as pipeline repair or the installation of hoses, in real time using an ROV (Remotely Operated Vehicle) simulation technology. This is a very useful system that gives a quick response [64].

The blockchain technology allowed the division into three groups, one formed by applications of its use in the sector, saving the industry from intermediaries and rebuilding all business models [65]. A second group was made up of publications aimed at strengthening security where it was stated that the blockchain technology allows extraction of the information of the contract directly, guaranteeing the reliability of the system. It also guarantees its authenticity and security, building a more ecological environment [66]. The last group was devoted to the search for energy efficiency from the perspective

of cryptocurrencies, because of the algorithms used for developing an increase of energy consumption, it was necessary to develop new algorithms [67].

With respect to the category of horizontal and vertical integration systems, the majority of publications addressed vertical integration. These publications aimed at the development of new products such as the aero-derivative gas turbine [68]. It is also used as an indicator of productivity in the sector, developing strategies that allow it to improve costs, quality indices, flexibility, and delivery time, among others [69], and is valued as an alternative in management by supporting efforts aligned with the supply chain and commercial strategy [70]. Other publications considered it to be positive compared to the transfer of information to foreign shipyards in an increasingly globalized world [71] and compared it to the alternative of outsourcing [72].

The groups into which we divided cybersecurity technology were focused on reducing environmental risks, so essential today that optical communications and quantum encryption are included to ensure the operations of the safest ships and to guarantee the safety of the oceans [73]. A second group studied the safety of the systems onboard by developing virtual laboratories to characterize and identify security events in maritime control systems [74]. Of course, there was a group aimed at supporting other technologies, as was the case with IoT. Considering that, as more devices are brought online, safety must be a major concern for users and operators. It is established that embedded applications should be built on a secure platform that can extend security features to the applications it houses [75]. Also included was a group aimed at general considerations of the implementation of the same [76].

Artificial intelligence is present in different aspects in the shipbuilding sector, although it was foreseeable that most of the publications would concern the development of improvements to navigation and control systems, such as algorithms that help route planning to avoid ship collisions [77]. In general, the technology has many applications in the sector from the design stages by studying the main dimensions, hull shape selection, stability or propulsion, through the use of artificial neural networks [78]. There was also a group of publications focused on decision support [79], energy efficiency, and even process optimization [80], such as the formation of certain parts of the hull by heating or mechanical forming by developing an automatic line heat-forming process based on the intensive application of numerical simulation and artificial intelligence [81].

The industrial Internet of Things (IoT), which allows field devices to communicate with each other and with the control systems with real-time responses, comprised three groups. The first group included the linkage it has with other technologies, as explained above with cybersecurity [82]. The second group included the use of technology to support the design stage of vessels, allowing for increased performance and value of the ship, although there are challenges to be considered in ensuring that relevant, accurate, and reliable data are articulated to stakeholders [83]. The third group was composed of publications that reflect the integration of processes and systems [84].

Within simulation, we distinguished, depending on the type, simulation by finite elements, simulation of discrete events, Smoothed Particle Hydrodynamics (SPH), Computational fluid dynamics (CFD). In the six established groups, we found studies with the different types of simulation previously identified. The first group contained the new propulsion systems in which electronic propulsion systems are established [85] or the effects that the hydrodynamic efficiency of the propellers has [86]. Both are physical simulations. The second group studied the ship structure and services with respect to crack propagation behavior [87] or residual stress analysis [88], finite element simulations. There were publications focused on spill prevention [89] and risk analysis due to high pressure of fuel gas in tanks [90]. The third group was dedicated to welding, which is so important in the shipbuilding sector [91] from the perspective of different positions [92] and deformations in assembly [93].

The next group [94] covered the study of navigation systems, and a fifth group [95] covered the supply chain. The sixth group included planning [96], production control [97] and optimization [98]. Finally, an important number of publications were dedicated to the earlier stages of shipbuilding, such as ship design as a tool for optimization [99] and analysis [100].

### 3.3. Evaluation of the Key Enabling Technologies according to Some Basic Categories

From the 12 Industry 4.0-enabling technologies specific to the shipbuilding sector, the categories that are common to most of them were identified. One of the common categories focused on the general study of technology in its application to the sector (66 studies). Another common approach with a larger number of articles focused on the application of systems improvement and efficiency, especially for navigation systems (66 studies). Another category was dedicated to design, including within this category tasks aimed at ship design and the design of new related products (64 studies). The other grouping subjects ranged from productivity, production control, decision support, process optimization, and the supply chain (16 studies). There was a specific category dedicated to one of the most relevant activities in the sector, which is the welding distributed in only two technologies: autonomous robots and simulation. (12 studies). This left other non-common categories with fewer than 10 studies (44 studies).

Finally, it was followed by two categories with the same number of studies, one dedicated to studying the effects on energy efficiency and environmental sustainability (16 studies). This was the indicator offered by the study of the trend followed by the scientific community in its approach to supply chain sustainability. It can be seen that less than 6% of the publications were of concern to researchers in the sector.

Not all KETs addressed the category of environment called Energy Efficiency and Environmental Sustainability. Only five of the 12 KETs dealt with studies related to sustainability. The weight fell on big data analytics with a total of nine of the 16 articles counted, the rest being distributed among cloud computing, blockchain, cybersecurity and artificial intelligence. It was very striking that the rest of the technologies did not incite any interest in this field, when it should include all technologies as being the key points of Industry 4.0. Figure 5 shows the seven categories addressed by each KET.

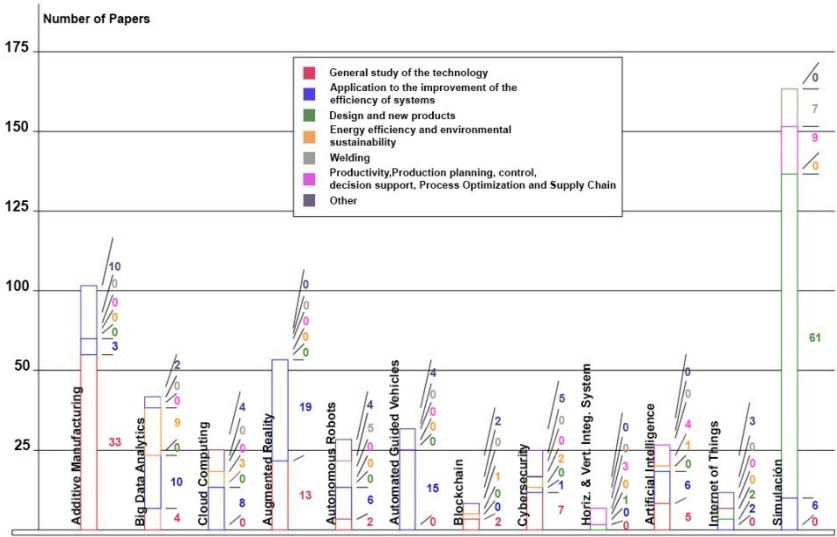

**Figure 5.** KET concerns distributed in the six basic categories.

This therefore highlighted the lack of integration of the environment into supply chain management, i.e., the economic, environmental, and social considerations necessary for the understanding of a sustainable supply chain are not taken into account. There is still a long way to go to implement sustainable practices and techniques for a sector such as shipbuilding, for which they carry considerable weight. This information should be taken into account both by the interested scientific community and by the companies of the sector, which besides being an imperative need, could be reversed as a competitive advantage.

## 4. Conclusions

The supply chain is considered today as a strategic tool because of its transversality, affecting all levels within an organization. One of the tasks that this study carried out was using it also as a guide to make the company more sustainable. Sustainability should be studied as a basic requirement in all sectors but it is more important for the shipbuilding industry. Such sustainability, carried out through the supply chain, will make the techniques and practices that are used applicable to different companies, thus contributing to the extension of an appropriate and extensible sustainability network. Industry 4.0 brings to the industry the necessity to implement new technologies that allow the industry to be updated, improving sustainability and benefits. This transformation also affects the supply chain in which, in addition to using new technologies, it considers the human and environmental dimensions. This is why the supply chain is now considered Sustainable Supply Chain 4.0.

Firstly, the enabling technologies of Industry 4.0 as adapted to the shipbuilding sector were studied. Studies of each one of them that belongs to the sector were located and grouped by categories, giving a clear vision of which technologies have been studied for the longest time and in which ones, it could be interpreted, the sector has not yet shown enough interest. Additionally, which countries are researching these technologies and which are not was studied at a global level, providing a clear vision of the major powers, such as the United States and Asia, that are choosing to advance in technologies such as additive manufacturing and simulation, which are experiencing a peak.

The proposed categories were measured in the literature and served as a starting point to study the implementation of supply chain model practices in shipbuilding. However, the research method carried out allows to have a general vision of what the scientific community is studying and what reference places are available to the companies that decide to consult about and improve their practices. One of the proposed categories is related to the environment, sustainability, and energy efficiency as key points for the advancement of technologies and coincides with one of the lines of interest of Industry 4.0. However, it was detected that not all KETs include in their research this key category, which one should emphasize above all: the big data analysis that presents studies focused on climate change and emissions reductions. Cloud computing is concerned with the subject, presenting articles focused on engine failure detection to reduce marine pollution and to reduce energy costs. Blockchain also shows interest in developing new algorithms that take into account this dimension. Cybersecurity deals with the security of ships, thus guaranteeing the oceans in the same way that artificial intelligence does.

With regard to simulation technology, it is important to add that although no direct reference was made, its indirect involvement with sustainability should be acknowledged. Moreover, it should also be noted that additive manufacturing did not have any articles either, despite the fact that it is considered to be a clean technology in its own right; it is an intrinsic property of the technology itself. However, this means that there is not enough attention paid to the subject and companies should provide the means to expand research in this area, so that they can implement sustainable policies at all levels, taking advantage of the transversality offered by the supply chain.

**Author Contributions:** M.R.-P. and M.B. conceptualized the paper. M.B. and F.J.A.F. approved the experimental procedure; M.R.-P. and J.S. analyzed the data; M.R.-P. wrote the paper; M.B. and J.S. revised the paper; F.J.A.F. supervised the paper. All authors have read and agreed to the published version of the manuscript.

**Funding:** This research received no external funding.

**Acknowledgments:** Navantia S.A. S.M.E. and University of Cadiz (UCA) supported this work.

**Conflicts of Interest:** The authors declare no conflict of interest.

# Appendix A

**Table A1.** Results categorized by key enabling technologies.

| KET | Interventions | Author(s) |
|---|---|---|
| Additive Manufacturing | Improvement techniques for the minimization of defects | [101] |
| | General study of the technology | [24–27,102–120] |
| | Parts manufacturing and repair | [28,29,121–128] |
| | Parts property improvement | [30,31,129–135] |
| | Redesign for application in additive manufacturing | [32,136,137] |
| Big Data Analytics | Process and system improvement | [33–35,138–144] |
| | Environmental studies | [36,37,80,145–150] |
| | Smart systems | [38,40] |
| | General study of the technology | [39,40,151,152] |
| Cloud Computing | Performance improvement in service | [41–43,153] |
| | Control system improvement | [44,45,47,154–158] |
| | Energetic efficiency and environmental sustainability | [46,47,158] |
| Augmented Reality | Learning and the influence of technology on the sector | [39,48,49,159–168] |
| | Simulated naval environments applied to navigation, safety and maintenance | [50–53,169–180] |
| | Application to the improvement of the efficiency of systems, mainly navigation | [54,181,182] |
| Autonomous Robots | Welding | [55,56,183–185] |
| | General study of the technology | [39,168] |
| | Improvement of system efficiency | [59,186] |
| | Cleaning, inspection and maintenance work | [57,58,187,188] |
| | Unmanned vehicles | [189–192] |
| Automated Guided Vehicle | System improvement | [60,193–196] |
| | Repairs, maintenance, and inspection | [61–64,197–202] |
| | Vehicle systems improvement | [203–206] |
| Blockchain | Applications of its use | [65,207] |
| | Strengthening security | [66,208] |
| | Energetic efficiency | [67] |
| Cybersecurity | General considerations in the implementation of the technology | [75,76,208–212] |
| | Environmental risk reduction | [73,213] |
| | Improving the safety of onboard systems | [74] |
| Horizontal and Vertical Integration System | New product development | [68] |
| | Impact on productivity | [69,214,215] |
| | Alternatives study | [70] |
| | Encouraging transfer | [216] |
| | Outsourcing comparison | [72,217,218] |

**Table A1.** *Cont.*

| KET | Interventions | Author(s) |
| --- | --- | --- |
| Artificial Intelligence | Navigation and control systems improvement | [77,219–223] |
| | General study of the technology | [39,78,82,224,225] |
| | Decision support | [79,226] |
| | Energy efficiency | [80] |
| | Process optimization | [81,227] |
| Internet of Things | Linking to other technologies | [49,82,228] |
| | Support to the design of ships | [83,225] |
| | Process and system integration | [84,168] |
| Simulation | New propulsion systems | [85,86,229] |
| | Structure and services ship study | [87–90,230–274] |
| | Welding | [55,91–93,275–277] |
| | Navigation systems study | [94,278,279] |
| | Supply chain | [95,280,281] |
| | Production planning and control | [96–98,282–284] |
| | Design | [40,99,100,285–293] |

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
