# Peer review of "Assessing Sustainability in the Shipbuilding Supply Chain 4.0: A Systematic Review"

_sustainability, doi:10.3390/su12166373_

Round 1
Reviewer 1 Report
The systematic literature review is interesting and the results are presented well. However, you could explain more carefully how these Key Enabling Technologies are identified (please add the references).
You have 289 references in your list: Would it be more unambiguous if the list of authors (table 1) would be added as an appendix divided as in table 1. ? Thus, references would include only those mentioned in the text.
The sustainability is “measured” (Chapter 3.3.) by the number of papers concerning energy efficiency and environmental sustainability. Figure 6 is a little difficult to read. Please use different colours and design.
Some points:
The title: The title of the paper is misleading: You do not study how sustainable the shipbuilding supply chain 4.0 is – just how “sustainability” is mentioned in the studies so far.
Introduction: This chapter is poorly written. You could explain what is Supply Chain 4.0 and how it is connected with “Sustainability”.
There are something missing: e.g. Figure 1: Methodology and Review (adapted from.?
Line 97 – Is the reference really (55)?
Please check the English!
Author Response
Thank you very much for your comments. In the attached file you can see our answers.

Reviewer 2 Report
Source of Figure 1 is missing.
Please provide the time information for the data analyzed across different countries.
The description of Figure 6 is too long. Please shorten it.
I would expect more discussions about the details in each category of the literature. For example, from line 200, the author discussed about blockchain technology and the subgroup of studies. More information of what’s going on in each subgroup should be provided. This is applied to the other topics as analyzed.
More discussions in the conclusion section about how the results are linked to sustainability should be made as the title is about sustainability.
The category of KET in Table 1 has been used by:
Ramirez-Peña, M.; Abad Fraga, F.J.; Sánchez Sotano, A.J.; Batista, M. Shipbuilding 4.0 Index Approaching Supply Chain. Materials 2019, 12, 4129.
Please justify how above mentioned paper fits into current paper.
Author Response

(The authors gave the same response as above.)

Reviewer 3 Report
Merci de m'avoir donné l'occasion de lire «La durabilité de la chaîne d'approvisionnement de la construction navale 4.0: un examen systématique».
Dans cet article, les auteurs cherchent à donner un aperçu général de l'état actuel de la construction navale adapté aux technologies clés de l'industrie 4.0.
Afin d'atteindre cet objectif, ils ont d'abord procédé à une revue systématique des travaux sur le terrain en divisant chacune des technologies en différentes catégories. Dans un deuxième temps, ils ont cherché à explorer la vision globale des pays intéressés par chacune des technologies habilitantes.
De cette façon, ils espèrent fournir une vision globale et encourager la recherche sur la «durabilité» de la chaîne d'approvisionnement de la construction navale.
Comme à la fin de l'introduction, cette étude vise à donner un aperçu de l'état actuel de la technique. Premièrement, il donne un aperçu de l'intérêt des pays pour les technologies clés génériques pour l'industrie 4.0. Deuxièmement, il donne un aperçu des études qui se concentrent sur la durabilité de la chaîne d'approvisionnement dans le but d'encourager une plus grande inquiétude à ce sujet.
L'idée du travail est intéressante et cible un domaine spécifique "la construction navale".
Cependant, pour améliorer le travail, certains points peuvent être discutés.
Dans l'ensemble du document la durabilité n'est traité que d'une façon très marginale. Le document, contrairement à son titre porte plus sur les industries clès de la construction navale ... la dirabilité (Limitée essentiellement dans le document à l'effeicacité énergétique) n'est pas soulevée que t
Is the concept of Industry 4.0 the same as that of key (enabling) technologies?
2. The choice of CIMO logic and PRISMA method was neither compared nor justified.
3. The title of figure N ° 1 to review
4.Attention to the refreneces. Example FOR reference 55 we do not see the correspondence between the paragraph and the source "ON BENDING PERFORMANCE OF ADDITIVELY MANUFACTURED STEEL CATENARY RISER (SCR): EFFECT OF WELDING RESIDUAL STRESS ON BENDING STRAIN CAPACITY"
5. China shows more interest in simulation (15/43) coinciding with South Korea (14/22), followed by Big Data (7/43) at lines 134 and 135. The wording of this sentence can cause confusion.
6. The document is more focused on key technologies than on the shipbuilding supply chain and even less than industry 4.0 and supply chain 4.0; Here we find the origin of a great contradiction with the tire proposed for the article.
7. Throughout the document sustainability is treated only in a very marginal way. The document, contrary to its title relates more to the key industries of the shipbuilding ... the dirability (Limited essentially in the document to energy efficiency) is raised only very rarely and in a very superficial way (not development, analysis pad, no explanation, no alternatives ...). In fact, the article focuses more on the ma, than interest in sustainability in the research work devoted to shipbuilding and not the opposite. In this sense it is rather judicious to review the title of the document to express this objective
8. Le rôle et l'objectif de la figure 6 ne sont pas clairs
Author Response

(The authors gave the same response as above.)

Round 2
Reviewer 1 Report
The corrections are made. Now the paper is better!
Author Response
Thank you very much.
Reviewer 2 Report
I have no further comment on the paper.
Author Response
Thank you very much.
Reviewer 3 Report
Une nette amélioration est notée dans le texte par rapport à la première version.
De plus, les auteurs ont répondu de manière satisfaisante aux différentes questions et remarques de la première relecture.
Trois petites remarques à prendre en considération:
1. L'annexe n ° 1 ne se trouve pas dans le texte.
2. L'auteur saute de 3.3 au point 5 (Conclusion). Le point 4 est oublié.
3. Attention, la bibliographie de la 1ère version est toujours visible sur le document à côté de la nouvelle bibliographie.

Author Response
Thank you for your comments. We have carefully reviewed the comments and have revised the manuscript accordingly.